24-epibrassinosteroid and jasmine oil improve vegetative growth and productivity of Flame Seedless grapevines under abiotic stresses

http://orcid.org/0000-0001-5154-5201 Alebidi Abdullah 1
Elaidy Ahmed A. 2
http://orcid.org/0000-0003-2109-0925 Abd El-Khalek Ahmed F. 2
Abd Elmaksoud Magda M. 3
http://orcid.org/0009-0005-4814-1368 Elmenofy Hayam M. 3
Elzainy Ahmed M. 2
Rihan Hail Z. 4
Abdel-Sattar Mahmoud 1 mmarzouk1@ksu.edu.sa
1 Department of Plant Production, College of Food and Agriculture Sciences, King Saud University , Riyadh , Saudi Arabia
2 Department of Horticulture, Faculty of Agriculture, Tanta University , Tanta , Egypt
3 Fruit Handling Research Department Horticulture Research Institute, Agricultural Research Center , Giza , Egypt
4 School of Biological and Marine Sciences/Plymouth, University of Plymouth , Plymouth , United Kingdom
Singh Anshuman
Electronic publication date: 2025 Oct 14
Publication date: 2025
Volume: 13
Electronic Location ID: e20181
Received 2025 Jul 7; Accepted 2025 Sep 12
Copyright: © 2025 Alebidi et al.
Copyright year: 2025
Copyright holder: Alebidi et al.
License: This is an open access article distributed under the terms of the Creative Commons Attribution License, which permits unrestricted use, distribution, reproduction and adaptation in any medium and for any purpose provided that it is properly attributed. For attribution, the original author(s), title, publication source (PeerJ) and either DOI or URL of the article must be cited.
License URL: https://creativecommons.org/licenses/by/4.0/

Keywords: Chemical characteristics, Enzyme activity, Fruit quality, Mineral content, Nutritional status, Physical characteristics, Yield

Funding: King Saud University, Riyadh, Saudi Arabia ORF-2025-707 This research was funded by the Ongoing Researcher Funding Program, (ORF-2025-707), King Saud University, Riyadh, Saudi Arabia. The funders had no role in study design, data collection and analysis, decision to publish, or preparation of the manuscript.

==============================
Abiotic stresses such as high temperature, humidity fluctuations, and excessive light negatively impact grapevine physiological functions, resulting in irregular vegetative growth and reduced productivity. Natural plant growth regulators and essential oils offer potential eco-friendly strategies to mitigate these adverse effects. This study investigated the effects of foliar applications of 24-epibrassinosteroid (Br) at concentrations of 1, 2, and 3 mg/L and jasmine oil (JO) at 500, 1,000, and 1,500 µL/L on Flame Seedless grapevines exposed to abiotic stress. The treatments aimed to enhance vegetative development, nutrient uptake, biochemical attributes, and yield. Results showed that all treatments successfully improved the vegetative growth of Flame Seedless grapevines by increasing leaf area, shoot length, diameter, number of leaves/shoots, pruning wood weight, internode length, and coefficient of wood ripening. They also improved the mineral content in leaf petioles, total carbohydrates in canes, chlorophyll contents in leaves, and yield per vine. In addition, the quality of the Flame Seedless grapevine was improved by increasing cluster weight, length, width, berry firmness, length, width, soluble solids content (SSC), titratable acidity (TA), SSC/TA ratio, total sugars, phenols, anthocyanin, and flavonoids, activities of peroxidase (POX), phenylalanine ammonialyase (PAL), polyphenoloxidase (PPO) and catalase (CAT) enzyme in berry. Application of Br at 3 mg/L yielded the highest significant values for vegetative growth parameters, yield, and physical characteristics. While JO at a rate of 1,500 µL/L increased the total phenols, flavonoids, and anthocyanin, as well as PPO, PAL, POX activity, and CAT in the berry. Foliar application of Br and JO effectively mitigated the adverse effects of abiotic stress in Flame Seedless grapevines.

Introduction

Grapes are in the genus Vitis, one of the sixteen genera of the Vitaceae family (Wen et al., 2018). Among the most significant and prevalent crops in the world is the grapevine (Chacón-Vozmediano et al., 2020). Red grapes are a valuable source for the food industry due to their high nutritional value, including organic acids, aromatic compounds, soluble sugars, vitamins, and antioxidants like phenolic compounds (anthocyanins, phenolic acids, stilbenes, and flavonols) and β-carotene (Abdel-Sattar et al., 2022). The Egyptian market places red grape varieties in a prominent marketing position, especially the Flame Seedless grape, one of the earliest seedless red grape varieties to ripen, and is highly sought after by consumers, as it closes a business gap in the market and is the most lucrative (Belal, El Kenawy & Omar, 2022).

Climatic conditions influence grapevine growth efficiency and crop phenology, with temperature being the primary driver of shifts in grapevine phenological stages (Cameron, Petrie & Barlow, 2022). For the grapevine, 25 °C to 35 °C is the ideal photosynthetic temperature (Kun et al., 2018). Heat acclimation mechanisms are triggered at temperatures above 35 °C, while physiological processes decrease at temperatures below 25 °C (Bernardo et al., 2018; Zhang et al., 2018). Because plants’ development rate is strongly temperature sensitive, a warmer climate will impair vegetative growth and productivity (Hatfield & Prueger, 2015). Long, hot, dry summers and brief, comparatively wet winters with moderate to high temperatures define the Mediterranean basin, one of the world’s greatest wine-growing regions (Venios et al., 2020). Due to the semi-arid climate, vineyards increasingly suffer from severe summer stress (Chacón-Vozmediano et al., 2020). Therefore, when the conditions necessary to meet developmental requirements are not suitable, specific disorders may occur that can be addressed through horticultural practices. These practices can be used to complete the vegetative growth and fruit metabolism stages, employing different approaches and methodologies, such as foliar spraying with brassinosteroids and jasmine oil.

Brassinosteroids (BRs) are a group of hormones with strong growth-regulating effects that function both independently and together with other phytohormones to control various BR-regulated activities (Zullo, 2018; Peres et al., 2019). 24-epibrassinolide is the most widely used BRs in research on the physiological effects of exogenous steroid phytohormones on plants (Fedina et al., 2017) because of its strong biological activity and extensive distribution (Bartwal et al., 2013). BRs serve critical and decisive functions in a wide range of growth and developmental responses during the plant life cycle (Manghwar et al., 2022), and they can create biological effects at extremely low concentrations (Abdel-Sattar et al., 2024). BRs regulate a variety of processes, including stem and root cell elongation and division, reproductive development, photomorphogenesis, leaf senescence, and stress response (Kim et al., 2012; Fariduddin et al., 2014; Wang, Yu & Xie, 2020). It can also contribute to higher chlorophyll content, enhanced photosynthesis efficiency, and the promotion of root development and seedling, blooming, and maturation, and it might be in charge of enhancing plants’ ability to withstand harsh environmental conditions like cold, drought, salt, and acid (Chen et al., 2017; El-Banna et al., 2022). Numerous physiological and molecular reactions in plants, including vascular differentiation, anthocyanin biosynthesis regulation, cell elongation, cell division, photomorphogenesis, gene expression, protein synthesis, and differentiation of numerous cell types, were demonstrated by BRs (Peres et al., 2019; Sun et al., 2020). Over the past few decades, plant scientists have become interested in BR research because of its adaptability in mitigating various environmental challenges (Divi, Rahman & Krishna, 2016; Manghwar et al., 2022). Furthermore, BR applied exogenously, or changes made to their production and signaling, may increase crop yields (Abdel-Sattar et al., 2024). Consequently, these substances can be applied to crops as biostimulants to increase plant efficiency and induce abiotic stress tolerance (Trevisan et al., 2020).

Recently, natural plant extracts have been explored as new alternatives to enhance plant productivity and fruit quality while serving as environmental and human safety agents (Shahbaz et al., 2022). Essential oils are increasingly recognized as natural antioxidants, making them a promising option for preserving agricultural crops (Yeamsuriyotai et al., 2025). Although the use of jasmine essential oil has been widely studied in many countries worldwide, research into the application of jasmine oil remains relatively limited (Phuc et al., 2019). Jasmine oil plays a role in stress prevention by biostimulating plant growth and yield, in addition, improving the physicochemical properties of the fruit (Ahmed et al., 2016; Prosche & Stappen, 2024). To our knowledge, there have only been a few reports on the influence of natural oils on plant development and yield, including the study by El-Tanany, El-Moghazy & Abdul-Aziz (2018) and Farouk et al. (2021), which demonstrated the possibility of employing essential oils as biostimulants to enhance plant growth and production. So, extra research is needed to examine the growth and yield of crops and the full biochemical characteristics of the effect of natural oils on crop productivity.

Grapevine productivity varies significantly between years due to climatic conditions. Given the projections of increased climate change, grape production, harvest quality, and longevity will be substantially affected. Therefore, this study seeks to stabilize productivity fluctuations and enhance the sustainability of grapevine supply in terms of quantity and quality using BR and JO under a semi-arid Mediterranean climate, particularly in Dakahlia Governorate, Egypt. So, the primary objective of this study is to investigate the effect of spraying BR and JO on improving grape growth and the physical and chemical quality of grape clusters in Flame Seedless grapes.

Materials and Methods

Plant materials and experimental procedure

During two seasons in 2023 and 2024, 10-year-old Flame Seedless grapevines grafted on Freedom grape rootstocks were studied in clay soil with a 1.5-m groundwater table and a flood irrigation system on the El-Baramon experimental farm in the Dakahlia Governorate (31°11′98′′ N, 31°45′13′′ E). Grapevines were cultivated using a quadrilateral cordon training system of short spurs in a Spanish baron trellis. All grapevines were spur pruned by the third week of January in both seasons, retaining 68 buds per vine (four cordons, each with three five-bud spurs and one two-bud renewal spur).

The irrigation supply was the Nile River, water samples were taken for analysis, the quality was determined, and referred to in Ali et al. (2014). Soil samples under each treatment were-collected from the root zone (0–90 cm) and assessed as a clay soil with an average of 7.3 pH, 1.06% organic material, 0.61 dS m−1 E.C., 1.7% CaCO3, 0.35 meq 100 g−1 HCO3−, 3.78 meq 100 g−1 SO42−, 1.0 meq 100 g−1Cl−, 0.85 meq 100 g−1 Na+, 0.75 meq 100 g−1 Ca2+, 0.82 meq 100 g−1 Mg2+, 24.6 mg kg−1 N, 16.0 mg kg−1 P, and 247.0 mg kg−1 K across seasons. The flood irrigation system was applied, and the irrigation water amount was based on climatic data collected from the nearest meteorological station. The applied irrigation requirement (IR) for each irrigation interval during the two growing seasons was calculated using the plant’s phonological stages and the percentage of growth shaded by the tree canopy. The total amount of water used was 1,090.9 m3/ha for each of the 11 irrigations over the season. During the growing season (March–October), the overall rate is around 12,000 m3/ha. Irrigation was done once a month during both seasons, except for May, June, and July, when it was done twice a month (Gaser, Abo El-Wafa & Abd El-Hameed, 2018). The control group consisted of grapevines at the study site under climate stresses, whose weather data are shown in Table 1, which received no treatments other than water spraying.

Table 1 Weather data from the El-Baramon experimental, Dakahlia Governorate, Egypt, from September 2022 to August 2024.

Month	Year	Temperature
(°C)	Humidity
(%)	Rainfall
(mm month−1)	Wind speed
(km h−1)	Cloud
(%)	Sun
(h month−1)	UV
index	
September	2022	32.2	58	0	12.4	5	376	7	
2023	29.5	56	0	13.5	5	375	7	
October	2022	29.5	60	0.8	11.3	10	386	6	
2023	31	64	0.3	12.1	11	386	5	
November	2022	25.8	67	2.6	11.4	8	375	6	
2023	27.1	62	1.8	10.6	27	356	6	
December	2022	17.7	71	4.2	13.4	22	377	5	
2023	19.5	70	5.3	10.5	19	367	4	
January	2023	15	59	4.1	14.2	31	366	4	
2024	19.5	64	3	13.3	19	382	5	
February	2023	16.2	62	10	12.5	31	331	4	
2024	19.7	65	4.6	12.8	27	321	4	
March	2023	18.5	63	2.3	14.9	23	368	7	
2024	21.3	67	6.7	14.5	14	391	5	
April	2023	21.9	56	0.8	13.6	15	379	7	
2024	27.2	56	3.8	15.6	9	383	8	
May	2023	27	43	0	14.5	10	390	7	
2024	37.9	42	0	13.6	3	395	8	
June	2023	30.6	49	0	13.8	7	383	8	
2024	37.3	46	0	13.2	1	383	9	
July	2023	34.7	51	0	12.8	8	395	8	
2024	37.7	46	0	13.5	2	395	9	
August	2023	34.1	52	0	13.1	4	395	8	
2024	34	48	0	12.1	2	395	8	

Jasmine essential oil (100%) was obtained from EL-Masrayia Company for Natural Oils in Cairo, Egypt, and stored in opaque bottles at 4 °C until used. GC-MS was used to identify the essential oil components at the National Research Center’s central laboratory in Giza, Egypt (Fig. 1). The source of brassinosteroid was 24-epibrassinosteroid, which was purchased from the Sigma Aldrich Company, St. Louis, MO, USA.

Figure 1 Chemical composition of jasmine oil (Jasminum grandiflorum L.) essential oil by GC–MS analysis.

Experimental design and treatments

This experiment included 63 grapevines as uniform as possible in vigor, canopy structure, and trunk diameter, planted at a spacing of 2 m × 3 m and free of any physiological diseases or nutrient deficiencies. The grapevines selected were foliar-sprayed with seven spray treatments, including the control treatment at the study site, whose weather data are shown in Table 1, which was sprayed with water. In both seasons, the same nine vines were subjected to the same treatment as follow: the control, distilled water-sprayed vines (T1), 24-epibrassinosteroid at 1 mg/L (T2), 24-epibrassinosteroid at 2 mg/L (T3), 24-epibrassinosteroid at 3 mg/L (T4), jasmine oil 500 µL/L (T5), jasmine oil 1,000 µL/L (T6) and jasmine oil 1,500 µL/L (T7). The vine had received ≈2.5 L until run-off at three different times: 2 weeks after the beginning of vegetative growth, when shoot length reached about 25–30 cm, after berry set, and the veraison stage.

Measurement and determinations

In mid-February during seasons 2023 and 2024, on each side of the vine, five non-fruiting shoots from the renewal spurs were selected at random and labeled to measure vegetative growth. The number of leaves per shoot, average shoot length using a measuring tape to the nearest centimeter (cm) in three separate readings and the average shoot diameter were calculated using a digital caliper. Two mature leaves on each marked shoot (i.e., the 6th and 7th from the shoot tip) were collected to measure leaf area (cm2) using a LI-3100 leaf area meter. The average leaf area was then determined. During dormant seasons, internode length (cm) was measured in five shoots per vine from the third base internode using a digital caliper with an accuracy of 0.01. Pruning products were weighed in kilograms per vine using a digital bench scale (Rotation Scales, Batavia, IL, USA) model PC-500. Wood ripening was recorded by labeling twelve shoots of the current season’s growth of each replicate to follow up on the average of wood ripening. According to Elsayed & El-Shewaikh (2023), five ripe cane samples were collected during the dormant season during the first week of November, and the coefficient of wood ripening was calculated by dividing the length of the ripened part of the shoot by the total shoot length.

The nutritional status of grapevines was assessed by determining the chlorophyll content of leaves, including chlorophyll a, b, and total chlorophyll, the mineral content of leaf petioles, and the cane total carbohydrate content. Two weeks following fruit set, samples of twenty fresh and mature apical fifth and sixth leaves from the leaves opposite the basal clusters on each shoot were collected from each side of the vine to determine the levels of chlorophyll a and b, as well as total chlorophyll, as defined by Lichtenthaler & Wellburn (1994). Chlorophyll b and chlorophyll a were measured at wavelengths of 646 and 663 nm, respectively, by a spectrophotometer (UV/Visible spectrophotometer, Libra SS0PC, Thermo Fisher Scientific, Waltham, MA, USA). Total chlorophyll contents (mg g−1 fw) were calculated using the following equations:

(1) Totalchlorophyll=[((chlorophylla+chlorophyllb)×extractvolume)/(1000×fw).

At dormant seasons, four non-fruiting shoots off the renewal spurs, two shoots at each side of the vine, were randomly selected by the end of the growing season in late December to assess total carbohydrates. Total carbohydrates as a percentage of dry weight were calculated using a spectrophotometer set (UV/Visible spectrophotometer, Libra SS0PC, Thermo Fisher Scientific, Waltham, MA, USA) at 490 nm. To determine macro- and micronutrient content, samples of 15 leaf petioles per replicate were dried at 60 °C for 72 h to constant weight. Following the pulverization of dried leaf petioles with the mortar and pestle equipment, concentrated sulfuric acid with repeated additions of 30% hydrogen peroxide was used to digest the powder (Wolf, 1982). Total nitrogen (g 100 g−1dw) and phosphorus (g 100 g−1dw) were measured colorimetrically using the generated solution and a spectrophotometer (UV/Visible spectrophotometer, Libra SS0PC, Thermo Fisher Scientific, Waltham, MA, USA) (Jones, Wolf & Mills, 1991). Potassium (g 100 g−1dw) was determined using the flame photometer (Tendon, 2005). Magnesium (mg g−1dw), calcium (g g−1dw), iron (mg g−1dw), manganese (mg g−1dw), and zinc (mg g−1dw) were determined using an atomic absorption spectrometer according to Carter (1993).

A sample of 27 clusters per treatment (three clusters/vine) was harvested in the second week of May during the 2023 and 2024 seasons, when the berries reached full color. Average yield/vine (cluster weight multiplied by the number of clusters/vine) was calculated for each treatment (kg/vine). To determine the physical qualities, ten clusters from each vine were randomly harvested. Each cluster was weighed individually, and the average cluster weight (g) was recorded. The average berry weight (g) was calculated by weighing 100 randomly selected berries from each cluster. Cluster length (cm) and width (cm) were measured from the uppermost berry to the lowest berry. A digital caliper with 0.01 mm accuracy was used to measure the diameter and length of the berries (mm). Berry firmness was measured and quantified in newtons (N) using a handheld digital penetrometer (FT-02) with a 2 mm plunger tip.

To determine the chemical properties of the berry juice, another random sample of vine berries was collected. At an air-conditioned room temperature (about 20–22 °C), the soluble solids content (°Brix) was measured using a handheld refractometer model N-1E (Atago Co., Ltd., Tokyo, Japan). Using phenol-phthalein as an indicator and NaOH (0.1 N), titratable acidity (TA) was estimated as a percentage of tartaric acid in 10 mL of juice (Association of Official Analytical Chemists (AOAC), 2019), and the SSC/TA ratio was calculated. The total sugars were measured colorimetrically using the phenol and sulfuric acid technique reported by Dubois et al. (1956). The spectrophotometer was used to measure the absorbance at 490 nm, and the concentration of total sugars was computed as g glucose 100/g fw (as a percentage).

The total anthocyanin, flavonoids, and total phenols content in berry skin (2 g) was determined using a methanolic HCL extraction solvent according to the method of Lee & Francis (1972) using a UV-Vis Spectrophotometer at wavelengths of 760, 510, and 535 nm, respectively. Total phenolic and total flavonoid contents were evaluated according to the protocol of Slinkard & Singleton (1977). The Folin-Ciocalteau reagent was used to measure a calibration curve for gallic acid concentrations to determine the total phenolic content as gallic acid equivalents (mg/100 g dry weight). A measurement of mg catechin equivalents/100 g (constant weight) was used to express total flavonoids by measuring a calibration curve for known catechin concentrations. Anthocyanin pigment was expressed as mg/100 g fresh weight.

The phenylalanine ammonia-lyase (PAL) contents of the grape juice extract were measured according to Jones (1984), using an aspectrophotometer at a wavelength of 290 nm at room temperature. Calculated PAL activity using the following equation:

PALactivity(Uming−1FW)=(ΔA290/6.22)×(1/30)×(1/1),

where ΔA290 is the change in absorbance at 290 nm, and 6.22 is the extinction coefficient of L-Phe. The peroxidase enzymatic activity was determined using a Hitachi U-2000 spectrophotometer (l = 460 nm) by the procedure outlined by Clemente & Pastore (1998). The polyphenoloxidase activity (l = 420 nm) was measured using the method described by Siddiq, Sinha & Cash, 1992. POX and PPO enzyme activity (U min g−1 FW). Catalase activity was determined using the method of Aebi (1984), and absorbance was measured at a wavelength of 240 nm using a spectrophotometer. Catalase activity was calculated using the following equation:

Catalaseactivity(Uming−1FW)=(ΔA240/0.0436)×(1/30)×(1/1),

where ΔA240 is the change in absorbance at 240 nm, and 0.0436 is the extinction coefficient of hydrogen peroxide according to Bergmeyer (1983). Units of catalase (U), which are the amount of enzyme needed to break down 1 μmol of hydrogen peroxide per minute at 25 °C and pH 7.0, are used to evaluate catalase activity.

Statistical analysis

Treatments were organized in a randomized complete block design (RCBD) system with three replicates, each replicate consisting of three vines. Data were first examined using the Shapiro-Wilk and Levene tests for normality and variance homogeneity, respectively. Before analyzing variance (ANOVA), the percentage data were first converted to the values of the Arcsine square root. The outcomes were then shown as back-transformed means. The ANOVA was performed using the CoStat program, version 6.311 (CoHort software, Monterey, CA, USA). Probability (p) < 0.05 was used for mean comparisons using Tukey’s honest significant difference (HSD) test. The score and loading plot for vegetative growth and biochemical parameters were generated using a principal component analysis (PCA) (Jolliffe, 2011). The two-way hierarchical cluster analysis (HCA) and heat map were generated using the means of the data matrices (Michie, 1982). Both PCA and HCA were performed using JMP Pro 16 (SAS Institute, Cary, NC, USA).

Results

Vegetative growth

Data presented in Table 2 indicate that spraying Flame Seedless grapevines three times with Br and JO significantly enhanced the vegetative growth characteristics as compared with the control. These characteristics include shoot length, diameter, leaf area, number of leaves/shoots, internode length (cm), pruning wood weight (Kg/vine), and coefficient of wood ripening during the 2023 and 2024 seasons. In both seasons, the highest significant values for the above traits were obtained by foliar with Br at 3 mg/L; the control recorded the lowest values. Data revealed non-significant differences between foliar JO at 1,000 and 1,500 µL/L on shoot length, leaf area, shoot diameter, number of leaves/shoot, internode length, coefficient of wood ripening, and pruning wood weight in both seasons of the study. No significant differences were found between foliar Br at 1 and 2 mg/L on shoot diameter, leaf area, and number of leaves per shoot during the 2024 season. Data also showed non-significant differences between foliar with Br at 1 and 2 mg/L on internode length and wood ripening in both seasons of the study.

Table 2 Effect of 24-epibrassinosteroid (Br) and jasmine oil (JO) on vegetative growth of Flame Seedless grapevines during the 2023 and 2024 seasons.

Season	Treatment	Shoot length (cm)	Shoot diameter (cm)	Leaf surface area (cm2)	Number of leaves per shoot	Internode length (cm)	Pruning wood weight
(Kg/vine)	Coefficient of wood ripening	
2023	Control	149.67 ± 2.31 e	1.12 ± 0.01 d	126.67 ± 1.44 e	19.90 ± 0.75 e	7.57 ± 0.19 e	1.65 ± 0.05 d	0.74 ± 0.01 e	
Br (1 mg/L)	167.67 ± 2.89 c	1.16 ± 0.03 cd	150.00 ± 1.90 b	22.37 ± 0.20 c	8.44 ± 0.11 c	2.29 ± 0.10 b	0.86 ± 0.01 b	
Br (2 mg/L)	175.67 ± 2.52 b	1.24 ± 0.04 ab	154.67 ± 1.83 b	23.47 ± 0.03 b	8.89 ± 0.07 b	2.29 ± 0.01 b	0.89 ± 0.01 ab	
Br (3 mg/L)	184.33 ± 2.51 a	1.29 ± 0.04 a	161.00 ± 1.28 a	24.54 ± 0.52 a	9.19 ± 0.04 a	2.51 ± 0.14 a	0.90 ± 0.02 a	
JO (500 µL/L)	157.00 ± 2.00 d	1.19 ± 0.01 bc	135.33 ± 2.01 d	20.90 ± 0.17 d	7.97 ± 0.15 de	1.96 ± 0.05 c	0.79 ± 0.01 d	
JO (1,000 µL/L)	162.33 ± 1.15 cd	1.22 ± 0.01 bc	140.00 ± 1.60 cd	21.60 ± 0.32 cd	8.17 ± 0.06 d	2.10 ± 0.08 bc	0.81 ± 0.01 cd	
JO (1,500 µL/L)	162.67 ± 1.53 c	1.25 ± 0.03 ab	142.33 ± 0.92 c	21.63 ± 0.10 cd	8.28 ± 0.07 d	2.30 ± 0.09 b	0.82 ± 0.01 c	
2024	Control	157.67 ± 5.77 e	1.17 ± 0.03 c	128.05 ± 1.75 f	20.97 ± 0.35 e	8.14 ± 0.08 e	1.82 ± 0.06 d	0.74 ± 0.02 d	
Br (1 mg/L)	175.67 ± 1.53 c	1.27 ± 0.02 ab	151.64 ± 0.61 c	23.40 ± 0.38 bc	8.90 ± 0.10 c	2.31 ± 0.05 b	0.87 ± 0.01 ab	
Br (2 mg/L)	178.33 ± 1.15 b	1.27 ± 0.02 ab	156.35 ± 1.41 b	23.82 ± 0.42 b	9.19 ± 0.05 b	2.45 ± 0.06 b	0.87 ± 0.01 ab	
Br (3 mg/L)	192.33 ± 2.16 a	1.33 ± 0.02 a	162.75 ± 1.24 a	25.62 ± 0.44 a	9.80 ± 0.10 a	2.65 ± 0.12 a	0.90 ± 0.01 a	
JO (500 µL/L)	162.33 ± 1.15 d	1.25 ± 0.02 b	136.81 ± 2.65 e	21.80 ± 0.30 de	8.51 ± 0.06 d	2.14 ± 0.08 c	0.83 ± 0.01 c	
JO (1,000 µL/L)	170.33 ± 3.21 cd	1.28 ± 0.01 ab	141.53 ± 2.80 ed	22.63 ± 0.17 cd	8.81 ± 0.11 c	2.25 ± 0.11 bc	0.84 ± 0.01 c	
JO (1,500 µL/L)	172.00 ± 1.00 c	1.25 ± 0.05 bc	143.88 ± 1.04 d	23.10 ± 0.21 bc	8.91 ± 0.10 c	2.19 ± 0.04 b	0.85 ± 0.01 bc	
Note:

Mean values within a column for each season, followed by different letters, are significantly different at p ≤ 0.05.

The nutritional status

After the application of Br and JO, there were statistically significant differences in the effect of all spraying via the leaves on the macro and micronutrient content of the Flame Seedless’ grapevines (p < 0.05). The results in Figs. 2A-2D and 3A-3D indicate that JO and Br treatments had a significant effect on P, N, Ca, K, Mg, Zn, Fe, and Mn contents in leaf petioles compared to untreated vines during both seasons. The data indicate that spraying Br was more effective than jasmine oil in improving the macroelements and microelements content in leaf petioles of vines. The greatest values of mineral contents were obtained due to spraying the vines with Br at 3 mg/L. Moreover, no significant differences were observed in the nitrogen, phosphorus, potassium, calcium, and magnesium contents of leaf petioles when Br was applied at concentrations of 1 and 2 mg/L during the two study seasons. Non-significant differences were also observed in the phosphorus content in leaf petioles in the two seasons, on the one hand, during the two study seasons, and the calcium content in the second year, on the other hand, when spraying with JO at concentrations of 500 and 1,000 µl/L. Moreover, no significant differences were observed between the 1,000 µL JO treatment and the 1 mg/L Br treatment for their effects on leaf petiole manganese and magnesium content during the first study season. In addition, no significant difference was observed between the 1 and 2 mg/L Br treatment and the 1,000 and 5,000 µL JO treatment on leaf petiole zinc content during the first study season.

Figure 2 Effect of the sprayed 24-epibrassinosteroid (Br) and jasmine oil (JO) on leaf contents of N (A), P (B), K (C), and Ca (D) of Flame Seedless grapevines during the 2023 and 2024 seasons.

The means with the same letters are insignificantly different at p ≤ 0.05 using Tukey’s HSD test.

Figure 3 Effect of the sprayed 24-epibrassinosteroid (Br) and jasmine oil (JO) on leaf contents of Mg(A), Fe (B), Zn (C), and Mn (D) of Flame Seedless grapevines during the 2023 and 2024 seasons.

The means with the same letters are insignificantly different at p ≤ 0.05 using Tukey’s HSD test.

The foliar application repercussions of JO and Br on chlorophyll a, chlorophyll b, and total chlorophyll contents in leaves, and total carbohydrates in cane contents of Flame Seedless grapevines grown in the 2023 and 2024 seasons are presented in Figs. 4A-4D. The foliar applications of JO and Br significantly improved leaf chlorophyll a, chlorophyll b, and total chlorophyll contents and cane carbohydrate content compared with the control. Foliar application with 3 mg/L Br was the most effective treatment in this study. Non-significant differences were found between foliar Br at 1 and 2 mg/L on leaf chlorophyll a, chlorophyll b, total chlorophyll contents, and cane carbohydrates content in both study seasons. Foliar spray data also showed no significant differences between different jasmine oil concentrations on total carbohydrates and chlorophyll a in the two seasons of the study, and different Br concentrations on chlorophyll a in the first season of the study. There were also no significant differences between spraying jasmine oil at concentrations of 1,000 and 1,500 µL/L on leaf chlorophyll b and total chlorophyll in the second season of the study.

Figure 4 Effect of the sprayed 24-epibrassinosteroid (Br) and jasmine oil (JO) on chlorophyll a (A), chlorophyll b (B), total chlorophyll content (C), and total carbohydrates (D) of Flame Seedless grapevines during the 2023 and 2024 seasons.

The means with the same letters are insignificantly different at p ≤ 0.05 using Tukey’s HSD test.

Total yield

Data on the influence of the two applied treatments, Br and JO, on the yield of Flame Seedless grapevines in both seasons are shown in Fig. 5. The data show that, in the 2023 and 2024 seasons, all applied treatments significantly improved yield per vine as compared to the control. The results revealed that the effect of Br concentrations was superior to that of jasmine oil concentrations on yield, which gradually improved with increasing Br and JO concentrations. After careful analysis of the data, it was found that, compared to all treatments in the seasons, the application of Br at 3 mg/L produced the highest significant yield/vine, while the control group had the lowest yield/vine. The data also showed no significant differences between JO at concentrations of 1,000 and 1,500 µl/L in vine productivity in both study seasons.

Figure 5 Effect of the sprayed 24-epibrassinosteroid (Br) and jasmine oil (JO) on yield per vine of Flame Seedless grapevines during the 2023 and 2024 seasons.

The means with the same letters are insignificantly different at p ≤ 0.05 using Tukey’s HSD test.

Physical characteristics

In the experiment, the physical characteristics of the clusters and berries in Flame Seedless grapevines changed in both seasons as a result of the foliar treatments of Br and JO (Table 3). The data show that, in the 2023 and 2024 seasons, all applied treatments significantly improved physical properties, including cluster weight, length, width, and weight of 100 berries per cluster, as well as berry length, berry width, and berry firmness, compared to the control in both seasons. As compared with the other treatments in both seasons, the physical characteristics increased at a Br dose of 3 mg/L in both experimental years. Also, the results revealed that the effect of Br concentrations is better than that of JO concentrations on the physical properties of the clusters and berries. There were no significant differences in cluster weight, length, width, and 100-grain weight per cluster between JO at concentrations of 1,000 and 1,500 µl/L in the first and second seasons. The data are also clear, with no significant differences between the Br treatments at concentrations of 2 and 3 mg/L in cluster weight during the two study seasons. Also, data showed that no significant differences between foliar Br at 1, 2, and 3 mg/L on berry firmness in the first season of the study.

Table 3 Effect of 24-epibrassinosteroid (Br) and jasmine oil (JO) on physical characteristics of Flame Seedless grapevines during the 2023 and 2024 seasons.

Season	Treatment	Cluster length (cm)	Cluster width (cm)	Cluster weight (g)	weight of 100 berries per cluster (g)	Berry length (mm)	Berry diameter (mm)	Berry firmness (N)	
2023	Control	21.00 ± 0.60 d	12.37 ± 0.39 c	469.33 ± 5.04 f	205.67 ± 3.16 e	14.67 ± 0.58 f	14.33 ± 0.58 c	1.95 ± 0.02 e	
Br (1 mg/L)	24.33 ± 0.58 bc	13.07 ± 0.31 bc	541.33 ± 11.02 c	231.00 ± 3.96 c	18.33 ± 0.58 bc	16.67 ± 0.58 ab	3.50 ± 0.03 a	
Br (2 mg/L)	25.67 ± 0.58 ab	13.97 ± 0.40 ab	566.00 ± 2.00 b	242.33 ± 3.11 b	19.00 ± 0.58 b	17.67 ± 0.58 a	3.62 ± 0.01 a	
Br (3 mg/L)	27.67 ± 0.58 a	14.57 ± 0.50 a	598.00 ± 8.98 a	261.00 ± 4.72 a	20.33 ± 0.58 a	17.67 ± 0.58 a	3.71 ± 0.03 a	
JO (500 µL/L)	21.37 ± 0.78 d	12.43 ± 0.12 c	495.33 ± 6.43 e	211.00 ± 3.56 ed	15.67 ± 0.58 ef	15.00 ± 1.00 bc	2.50 ± 0.02 d	
JO (1,000 µL/L)	22.37 ± 1.18 cd	12.73 ± 0.06 c	505.33 ± 5.03 de	219.00 ± 3.00 d	16.33 ± 0.58 de	15.33 ± 0.58 bc	2.86 ± 0.02 c	
JO (1,500 µL/L)	23.00 ± 1.00 cd	12.9.0 ± 0.20 c	521.00 ± 7.00 cd	223.00 ± 4.46 cd	17.33 ± 0.58 cd	16.00 ± 0.10 abc	3.24 ± 0.01 b	
2024	Control	20.57 ± 0.51 f	12.67 ± 0.31 c	451.67 ± 2.31 f	207.70 ± 3.20 f	14.67 ± 0.58 c	15.67 ± 0.58 c	2.33 ± 0.15 f	
Br (1 mg/L)	24.53 ± 0.40 c	14.67 ± 0.49 ab	531.67 ± 8.08 c	233.28 ± 2.07 c	17.67 ± 0.58 ab	18.00 ± 1.00 a	3.84 ± 0.10 bc	
Br (2 mg/L)	25.83 ± 0.40 b	15.47 ± 0.38 a	560.33 ± 9.02 b	244.73 ± 3.17 b	17.33 ± 0.58 ab	18.67 ± 0.58 a	3.94 ± 0.07 b	
Br (3 mg/L)	27.53 ± 0.70 a	15.77 ± 0.40 a	597.00 ± 7.07 a	263.58 ± 4.80 a	17.67 ± 0.58 a	19.00 ± 0.10 a	4.16 ± 0.05 a	
JO (500 µL/L)	21.83 ± 0.55 e	13.53 ± 0.25 bc	473.67 ± 6.02 e	213.08 ± 4.02 df	15.33 ± 0.58 bc	17.00 ± 0.10 bc	2.95 ± 0.01 e	
JO (1,000 µL/L)	22.53 ± 0.21 de	14.00 ± 0.20 b	489.00 ± 5.29 de	221.16 ± 3.52 d	15.67 ± 0.58 bc	17.00 ± 1.00 bc	3.58 ± 0.08 d	
JO (1,500 µL/L)	23.50 ± 0.52 cd	13.90 ± 0.53 b	509.67 ± 7.55 d	225.20 ± 3.50 d	16.67 ± 0.58 ab	17.67 ± 0.58 ab	3.72 ± 0.11 cd	
Note:

Mean values within a column for each season that are followed by different letters are significantly different at p ≤ 0.05.

Chemical characteristics

The data shown in Table 4 revealed that SSC, SSC/TA ratio, total sugars, total phenols, total anthocyanin, and flavonoids were improved gradually while decreasing acidity in both seasons, with increasing concentrations of JO and Br compared to the control. Also, the results revealed that the effect of JO concentrations is better than that of Br concentrations on the TA, SSC, SSC/TA ratio, total sugars, total anthocyanin, total phenols, and flavonoids of berries. According to additional data analysis, the application of JO at 1,500 µL/L in this study was the most effective in raising the SSC, SSC/TA ratio, total sugars and decreasing TA in comparison to the other treatment, while control reduced SSC, SSC/TA ratio and total sugars and increasing TA. Also, data showed that no significant differences between foliar JO at 500, 1,000, and 1,500 µL/L on total phenols in both seasons of the study.

Table 4 Effect of 24-epibrassinosteroid (Br) and jasmine oil (JO) on chemical characteristics of Flame Seedless grapevines during the 2023 and 2024 seasons.

Season	Treatment	SSC (°Brix)	TA (%)	SSC/TA ratio	Total sugars (%)	Total anthocyanins
(mg/g fw)	Total phenols
(mg/100 g fw)	Flavonoids
(mg/100 g fw)	
2023	Control	16.60 ± 0.15 f	0.66 ± 0.01 a	24.97 ± 0.28 c	13.66 ± 0.09 f	29.83 ± 0.20 f	40.84 ± 0.40 c	13.77 ± 0.71 d	
Br (1 mg/L)	17.07 ± 0.12 e	0.64 ± 0.00 b	26.69 ± 0.09 c	14.04 ± 0.15 e	30.31 ± 0.10 ef	42.47 ± 0.56 bc	14.74 ± 0.50 d	
Br (2 mg/L)	17.33 ± 0.12 de	0.62 ± 0.00 c	28.17 ± 0.09 c	14.36 ± 0.09 de	30.85 ± 0.32 de	44.31 ± 0.82 b	14.87 ± 0.03 cd	
Br (3 mg/L)	17.47 ± 0.12 d	0.59 ± 0.01 c	29.37 ± 0.09 ab	14.53 ± 0.06 d	31.18 ± 0.14 cd	44.35 ± 0.96 b	16.70 ± 0.51 b	
JO (500 µL/L)	17.93 ± 0.12 c	0.56 ± 0.01 d	32.29 ± 0.09 bc	14.69 ± 0.05 c	31.79 ± 0.19 bc	59.01 ± 0.31 a	16.28 ± 0.45 bc	
JO (1,000 µL/L)	18.27 ± 0.23 b	0.53 ± 0.01 e	34.64 ± 0.18 bc	14.90 ± 0.07 b	32.45 ± 0.31 b	60.73 ± 0.64 a	16.99 ± 0.45 b	
JO (1,500 µL/L)	18.73 ± 0.12 a	0.48 ± 0.00 f	38.81 ± 0.09 a	15.23 ± 0.11 a	33.34 ± 0.19 a	61.53 ± 0.08 a	18.51 ± 0.54 a	
2024	Control	16.87 ± 0.13 f	0.64 ± 0.01 a	26.52 ± 0.40 d	13.95 ± 0.11 f	30.21 ± 0.15 e	41.24 ± 0.40 c	13.27 ± 0.50 e	
Br (1mg/L)	17.33 ± 0.23 e	0.58 ± 0.02 bc	29.85 ± 1.80 c	14.33 ± 0.11 e	30.98 ± 0.11 d	42.88 ± 0.57 bc	14.46 ± 0.47 de	
Br (2mg/L)	17.73 ± 0.12 d	0.54 ± 0.00 c	32.80 ± 0.60 bc	14.67 ± 0.10 d	31.79 ± 0.02 c	44.75 ± 0.83 b	15.00 ± 0.01 cd	
Br (3mg/L)	17.93 ± 0.12 cd	0.49 ± 0.01 d	36.75 ± 1.06 a	14.83 ± 0.10 cd	32.12 ± 0.22 bc	44.79 ± 0.97 b	16.54 ± 0.73 c	
JO (500 µL/L)	18.13 ± 0.12 c	0.60 ± 0.02 ab	30.36 ± 2.18 bc	15.00 ± 0.10 c	32.61 ± 0.32 b	59.59 ± 0.30 a	16.45 ± 0.45 c	
JO (1,000 µL/L)	18.40 ± 0.20 b	0.55 ± 0.01 c	33.41 ± 1.54 b	15.22 ± 0.08 b	33.27 ± 0.21 a	61.33 ± 0.64 a	19.03 ± 0.50 b	
JO (1,500 µL/L)	18.80 ± 0.20 a	0.49 ± 0.01 d	38.37 ± 1.03 a	15.55 ± 0.11 a	33.69 ± 0.12 a	61.93 ± 0.31 a	20.74 ± 0.63 a	
Note:

Mean values within a column for each season that are followed by different letters are significantly different at p ≤ 0.05.

Enzyme activity

The data shown in Figs. 6A–6D indicate that the two treatments used, Br and JO, affected the POX, PAL, PPO, and CAT activities of Flame Seedless grapevines throughout both seasons. According to additional data analysis, the application of JO at 1,500 µL/L in this study was the most effective in raising the POX, PAL, PPO, and CAT activities in comparison to the other treatments, while the control reduced POX, PAL, PPO, and CAT activities in both seasons. Also, data showed that non-significant differences between foliar jasmine oil at 1,000 and 1,500 µL/L on POX activity in the both seasons of study and data showed that non-significant differences between foliar Br at 1 and 2 mg/L on POX activity and pectin methyl in the two seasons of study and data showed that no significant differences between foliar Br at 2 and 3 mg/L on PAL enzyme activities.

Figure 6 Effect of the sprayed 24-epibrassinosteroid (Br) and jasmine oil (JO) on activities of POX (A), PAL (B), PPO (C), and CAT (D) enzymes of Flame Seedless grapevines during the 2023 and 2024 seasons.

The means with the same letters are insignificantly different at p ≤ 0.05 using Tukey’s HSD test.

Principal component analysis and hierarchical cluster analysis

The purpose of applying PCA and HCA was to give a more comprehensive image of the sprayed Br and JO on vegetative growth parameters and biochemical attributes of Flame Seedless grapevines during the 2023 and 2024 seasons. As for the PCA (Fig. 7), the PCA score plots demonstrated a clear separation among treatments, with samples treated with Br (1, 2 and 3 mg/L) distinctly clustering on the positive side of the first principal component (PC1), reflecting their substantial positive influence on most measured traits in both seasons (Figs. 7A and 7C). The first two principal components accounted for 64.5% and 29.0% of the total variance in 2023, and 67.4% and 24.7% in 2024, respectively, confirming the efficiency of PCA in capturing the major sources of variability. Treatments with JO at 500, 1,000, and 1,500 µL/L were positioned intermediately between Br and the control. While the control treatment consistently aligned on the negative side of PC1, indicating comparatively lower effectiveness.

Figure 7 Principal component analysis (PCA) showing the score plots (A, C) and loading plots (B, D) of sprayed 24-epibrassinosteroid (Br) and jasmine oil (JO) on vegetative growth parameters and biochemical attributes of Flame Seedless grapevines during the 2023.

Values are the means of three replicates (n = 3).

Across both seasons, Br treatments were strongly associated with the positive side of PC1, reflecting their pronounced positive impact on parameters such as shoot diameter, pruning weight, total chlorophyll content, chlorophyll a and b, leaf surface area, cluster weight, and concentrations of essential macro- and micronutrients (N, P, K, Ca, Mg, Fe, Zn, Mn). Meanwhile, the control treatment consistently clustered on the negative side of PC1, corresponding to reduced performance in most studied traits. Notably, variables such as total anthocyanins, total flavonoids, total phenols, and total sugars contributed substantially to the variation along PC2 in both seasons, further discriminating between treatments (Figs. 7B and 7D).

Complementarily, HCA (Figs. 8A and 8B) accompanied by heatmap visualization further validated these findings. The highest and lowest values are represented with blue and white colors, respectively. The heatmaps revealed that Br treatments exhibited the highest values across most evaluated physiological and biochemical traits in both seasons. JO treatments showed moderate values, whereas the control consistently presented the lowest values. This clustering pattern confirmed the differentiation observed in PCA, emphasizing the superior efficacy of Br in enhancing grapevine growth and biochemical performance.

Figure 8 Two-way hierarchical cluster analysis (HCA) and heat map showing the effect of sprayed 24-epibrassinosteroid (Br) and jasmine oil (JO) on vegetative growth parameters and biochemical attributes of Flame Seedless grapevines during the 2023 (A) and 2024 (B).

Values are the means of three replicates (n = 3).

Discussion

Grape production in the Mediterranean region faces several abiotic stresses, including high humidity, extreme temperatures, and intense sunlight. These climatic conditions, shaped by North Africa’s tropical influence, negatively affect grape quality and yield, resulting in inconsistent productivity (Greer & Weston, 2010; Kun et al., 2018). Stress induces reactive oxygen species that cause membrane lipid peroxidation, disrupting physiological processes such as water retention, nutrient balance, and the uptake of essential macro- and micronutrients (Abdel Samad & Shaaban, 2024). Foliar spraying with JO and Br on grapevines exposed to heat stress significantly improved vegetative growth (Table 2) and consequently, enhanced nutrient uptake, internode length, pruning wood weight, and the coefficient of wood ripening. This, in turn, enhanced the plant’s ability to withstand environmental stresses and reduced ROS (Sabry, El-Helw & Abd El-Rahman, 2011; Ahammed et al., 2020).

These findings might be explained by the significance of plant extract oil’s function in enhancing numerous physiological and biochemical processes by stimulating meristem tissue, promoting cell division and elongation, and thereby enhancing all growth characteristics (Sabry, El-Helw & Abd El-Rahman, 2011; Mahmoud et al., 2024). Jasmine oil extract significantly improves vegetative growth parameters due to its content of benzyl acetate, Cis-jasmine, Methylanthranilate, Geraniol, Benzyl benzoate, Phytol, Linalool, indole, Carvacrol, α-pinene, Limonene, and octen-3-ol (Ahmed et al., 2016; Prosche & Stappen, 2024). It also maintains a larger leaf area for restored photosynthetic activity, encouraging plants to gradually accumulate beneficial elements. This is due to the important role that sugar alcohols such as sorbitol and mannitol play in improving nutrient mobility within plants by helping their transport over long distances through the phloem (Mahmoud et al., 2024). The increase in vegetative characteristics may be attributed to Br, which promotes carbohydrate assimilation, photosynthesis, cell division and elongation, protein, and nucleic acid synthesis (Asghari & Rezaei-Rad, 2018; Senthilkumar, Vijayakumar & Soorianathasundaram, 2018; Tanveer et al., 2019; Ahammed et al., 2020). This, in turn, improves the percentage of total carbohydrates in canes, pruning wood, weight, and ripening wood.

Exogenous JO and Br foliar addition in hot-stressed grapevine helps to improve macronutrients and micronutrients, chlorophyll, and total carbohydrates content (Figs. 2, 3, 4) by ROS removal, counteracting oxidative stress (Wang et al., 2018; Nazir et al., 2023), limiting lipid peroxidation of lipids and cell damage (Ahanger et al., 2023; Abdel-Sattar et al., 2024). The positive effects of JO and Br on improving vegetative growth and mineral nutrients help overcome abiotic stresses, ultimately increasing the total yield of the vineyard (Fig. 5) by improving cluster length, width, and weight. These natural oils may increase yield by stimulating specific physio-biochemical processes, such as enhancing photosynthetic rate, chlorophyll content, and sugar accumulation, thereby improving the efficiency of photoassimilate production and its translocation to fruiting organs (Farouk et al., 2021). Also, the main reason for the positive effects of essential oils on fruit yield may be due to the enhancement of potassium content, which accelerates photosynthesis processes and hence biomass production (Muetasam et al., 2022). Furthermore, using essential oils may enhance productivity by improving pollen grain germination and viability, as well as enhancing pollination processes by lengthening pollen tubes (Farouk et al., 2021). Additionally, essential oil application lowers ethylene production, increasing fruit yield per plant (Fig. 5) (Duque et al., 2021). The positive influence of brassinosteroids on improving yield may refer to their important role in promoting photosynthesis, carbohydrate assimilation, cell division, and elongation (Tanveer et al., 2019). BRs provide plants with resistance to biotic and abiotic stressors, resulting in significant improvements in physiological processes ranging from flower opening to fertilization and fruit set, with positive impacts on yield (Ahammed et al., 2020). In addition, BRs have a significant impact on fruit yield at harvest because of their extraordinary effect on photosynthetic carbon absorption efficiency, which promotes fruit set while decreasing fruit abscission and delayed senescence (Işçi & Gökbayrak, 2015).

The different treatments in both seasons improved the physical characteristics of Flame Seedless grapevines (Table 3). The physical properties are affected as a result of the activation of photosynthesis within the vine canopy through increased light penetration, accompanied by an enhancement in sugars in the berries, which increases their osmotic pressure and attraction force of water, leading to improved physical properties of the berries and clusters (El-Tanany, El-Moghazy & Abdul-Aziz, 2018; Garrido et al., 2018). Additionally, the growth produced by Br and JO as a result of increased cellular development during cell elongation and cell division due to their loosening impact on the cell wall also improved carbohydrate supply and reduced stress circumstances, which enhanced fruit physical properties (Ghorbani, Eshghi & Haghi, 2017; Farouk et al., 2021; Prosche & Stappen, 2024). The beneficial impact of Br treatments on enhancing the firmness of berries may be due to increased Ca2+, protopectin, and pectin in cell walls (Sharma, 2021).

Data presented in Table 4 indicated that spraying vines with Br and JO applications improved the chemical characteristics (Table 4). As affected by Br and JO, the increase of TSS could be attributed to alleviating the effect of high ambient temperatures and translocating sugars from leaves to fruits, or the transport of photo-assimilates to the fruit via the phloem (Sabry, El-Helw & Abd El-Rahman, 2011; Wang et al., 2019; Li et al., 2021; Abd El-Baset & ElMongy, 2023). A higher rate of photosynthesis may have led to increased carbohydrate accumulation in fruits, which would explain the improvement in total sugars (Abdel-Sattar et al., 2024). The increased sugar content of grapes is attributed to the overexpression of hexose transporter genes resulting from applying Br or JO (Belal, El Kenawy & Omar, 2022; Abd El-Baset & ElMongy, 2023). Plant tolerance for abiotic stress can be triggered by phenolic substances, such as flavonoids and anthocyanins, secondary metabolites, which can mediate the detrimental ROS scavenging by promoting the phenylpropanoid pathway and redox homeostasis (Zafari et al., 2020; Ramadan et al., 2024). It is suggested that external applications of jasmine oil and brassinosteroids affect enzymes and genes involved in the biosynthesis of phenolic compounds, acting as a signaling molecule and stimulating the accumulation of secondary metabolites, which was reflected in the improvement of phenolic compounds (Bartwal et al., 2013; Khetsha et al., 2022).

The activities of peroxidase (POX), phenylalanine ammonia-lyase (PAL), polyphenol oxidase (PPO), and catalase (CAT) enzymes were significantly affected by treatments (Fig. 6). Under normal circumstances, cells protect themselves from ROS damage by keeping ROS low through the activity of several antioxidant enzymes (Foyer & Noctor, 2005). Thus, many plant species depend on high amounts of POX, PAL, PPO, and CAT to survive environmental stresses (Deng et al., 2018). Phenylalanine ammonia-lyase (PAL) is considered a key enzyme in the regulation of anthocyanin accumulation during fruit ripening and maturation (Xi et al., 2013), as it catalyzes the first step in the phenylpropanoid pathway, leading to the production of anthocyanin precursors (Winkel-Shirley, 2002). In contrast, polyphenol oxidase (PPO) contributes to the formation of certain pigments, such as browning-related compounds, and plays a crucial role in secondary metabolism, particularly in the oxidation of phenolic compounds (Araji et al., 2014; Sullivan, 2015). POX and CAT are regarded as both protective and defensive enzymes that catalyze the oxidation of a variety of phenolic compounds while also consuming H2O2 and synthesizing lignin and suberin (Deng et al., 2018). Some specialized pigments are enhanced by stimulating color-related enzymes such as PPO and PAL, which play a new and essential role in secondary metabolism. The process of increasing red pigmentation occurs by stimulating anthocyanin-associated enzymes, such as PAL, which play a crucial role in the anthocyanin biosynthesis pathway. The increased activity of enzymes linked to secondary metabolism, the generation of secondary metabolites, and the ensuing scavenging of reactive oxygen species (ROS) are the reasons why BR therapy raises the activity of PAL and PPO (Xi et al., 2013; Asghari & Rezaei-Rad, 2018).

Conclusions

The current study concluded that spraying of 24-epibrassinosteroid (Br) and jasmine oil (JO) showed efficacy in alleviating the detrimental impacts of abiotic stressors on Flame Seedless grapevines. Both treatments markedly improved vegetative growth, nutritional composition, photosynthetic pigments, yield, and fruit quality. Br at 3 mg/L was more beneficial in enhancing vegetative growth and increasing physical fruit characteristics. Still, JO at 1,500 µL/L was more impactful in augmenting the biochemical composition of the berries, especially with antioxidant-related substances and enzyme activities. These results endorse the possibility of Br and JO as natural, environmentally sustainable agents for enhancing grapevine performance under adverse environmental circumstances. This information may be useful in developing new alternatives for natural plant extracts to reduce great economic losses for farmers under semi-arid climates. However, the roles of certain hormones, especially brassinosteroids, in meeting developmental requirements due to the semi-arid climate, are not fully understood and need further investigation.

Supplemental Information

Supplemental Information 1 The raw measurements are available in the supplementary file 1.

Additional Information and Declarations

Competing Interests

The authors declare that they have no competing interests.

Author Contributions

Abdullah Alebidi conceived and designed the experiments, performed the experiments, prepared figures and/or tables, authored or reviewed drafts of the article, and approved the final draft.

Ahmed A. Elaidy conceived and designed the experiments, performed the experiments, authored or reviewed drafts of the article, and approved the final draft.

Ahmed F. Abd El-Khalek conceived and designed the experiments, performed the experiments, analyzed the data, prepared figures and/or tables, authored or reviewed drafts of the article, and approved the final draft.

Magda M. Abd Elmaksoud conceived and designed the experiments, performed the experiments, analyzed the data, authored or reviewed drafts of the article, and approved the final draft.

Hayam M. Elmenofy conceived and designed the experiments, performed the experiments, analyzed the data, authored or reviewed drafts of the article, and approved the final draft.

Ahmed M. Elzainy conceived and designed the experiments, performed the experiments, analyzed the data, authored or reviewed drafts of the article, and approved the final draft.

Hail Z. Rihan conceived and designed the experiments, authored or reviewed drafts of the article, and approved the final draft.

Mahmoud Abdel-Sattar conceived and designed the experiments, performed the experiments, analyzed the data, prepared figures and/or tables, authored or reviewed drafts of the article, project administration, and approved the final draft.

Data Availability

The following information was supplied regarding data availability:

The raw measurements are available in the Supplemental File.

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
