# Peer review of "-epibrassinosteroid and jasmine oil improve vegetative growth and productivity of Flame Seedless grapevines under abiotic stresses"

_PeerJ, doi:10.7717/peerj.20181_

## Round 0.1 · original submission · Major Revisions

· Academic Editor

Major Revisions

Dear Dr. Abdel-Sattar

Thank you for your submission to PeerJ.

It is my opinion as the Academic Editor for your article - 24-epibrassinosteroid and jasmine oil improve vegetative growth and productivity of flame seedless grapevines under abiotic stresses - that it requires various major and minor revisions. You are therefore advised to go through the reviewers' comments, and modify your manuscript as per their suggestions.

It is pertinent to mention that your revised submission will undergo additional peer review to ensure that it is suitable for publication.

Hope to receive the revised submission in due course.

**Language Note:** The review process has identified that the English language must be improved. PeerJ can provide language editing services - please contact us at [email protected] for pricing (be sure to provide your manuscript number and title). Alternatively, you should make your own arrangements to improve the language quality and provide details in your response letter. – PeerJ Staff

Reviewer 1 ·

Basic reporting

The manuscript may be considered for acceptance after minor revision.

Experimental design

The manuscript contains several errors in the Materials and Methods section as well as in the Discussion part. The authors are advised to address these issues, which have been highlighted in the revised manuscript.

Validity of the findings

Needs improvement

Annotated reviews are not available for download in order to protect the identity of reviewers who chose to remain anonymous.

Reviewer 2 ·

Basic reporting

The manuscript title is vague since abiotic stress is very broad area, which particular stresses are targeted . So, only those relevant to the study should be reflected in the title. Accordingly, it should be modified. and should also include the effect of theses treatment on some selected abiotic stress parameters
However the article included sufficient introduction and background

Experimental design

The major weather parameters of the study area need to be provided
The major features of the study soil should be reported
Under the head ‘Measurement and Determinations’ there should be separate sub-sections each dealing with different group of observations.
The methodologies presented for various biochemical parameters should be crystal clear for the sake of reproducibility. Various reagents and instruments should be followed by their providers/companies with model and country of origin.

Validity of the findings

Authors are advised to include some additional data analysis such as correlation and PCA and elaborate their results to make the findings more attractive to the prospective readers.
Tables and figures should not be mentioned in the Discussion section
The conclusions section essentially reiterates the findings, and fails to capture the key messages for other researchers engaged in similar lines of work. It also does not mention the future course of action, highlighting what the present study could not achieve and what needs to be additionally assessed. Therefore, it should be critically revised to convey these points

Additional comments

Overall, the English language also needs improvement. The recommendation is to thoroughly revise the manuscript and resubmit it for further evaluation.

Annotated reviews are not available for download in order to protect the identity of reviewers who chose to remain anonymous.

Reviewer 3 ·

Basic reporting

This manuscript presents a well-executed study addressing the impact of 24-eBL and jasmine oil on the vegetative growth, yield and fruit quality of Flame Seedless grapes under abiotic stress conditions. The paper is well-structured, with clearly defined objectives, and the results are logically interpreted within the context of existing literature. The figures and tables are well-organized and effectively support the data; however, slight improvements in figure resolution may be necessary for publication. The discussion is comprehensive and connects the physiological and biochemical mechanisms behind the observed effects, providing a clear scientific explanation of the benefits of both treatments.
In general, the manuscript represents a valuable contribution to the field of horticulture and plant stress physiology. With only minor revisions, I believe this manuscript is suitable for publication in PeerJ.
Minor points
Line 146 the phrase "1000 cm tape" is unclear. If the authors are referring to a measuring tape or a ruler used to measure shoot length, it would be more appropriate to simply state "a measuring tape" or "a ruler."
Line 192 The phrase "10 milliliters" should be replaced with "10 mL" to conform to standard scientific conventions for units.
Line 197 The phrase "2 grams" should be replaced with "2 g" to conform to standard scientific conventions for units.

Experimental design

This manuscript presents a well-executed study addressing the impact of 24-eBL and jasmine oil on the vegetative growth, yield and fruit quality of Flame Seedless grapes under abiotic stress conditions. The paper is well-structured, with clearly defined objectives, and the results are logically interpreted within the context of existing literature. The figures and tables are well-organized and effectively support the data; however, slight improvements in figure resolution may be necessary for publication. The discussion is comprehensive and connects the physiological and biochemical mechanisms behind the observed effects, providing a clear scientific explanation of the benefits of both treatments.
In general, the manuscript represents a valuable contribution to the field of horticulture and plant stress physiology. With only minor revisions, I believe this manuscript is suitable for publication in PeerJ.
Minor points
Line 146 the phrase "1000 cm tape" is unclear. If the authors are referring to a measuring tape or a ruler used to measure shoot length, it would be more appropriate to simply state "a measuring tape" or "a ruler."
Line 192 The phrase "10 milliliters" should be replaced with "10 mL" to conform to standard scientific conventions for units.
Line 197 The phrase "2 grams" should be replaced with "2 g" to conform to standard scientific conventions for units.

Validity of the findings

This manuscript presents a well-executed study addressing the impact of 24-eBL and jasmine oil on the vegetative growth, yield and fruit quality of Flame Seedless grapes under abiotic stress conditions. The paper is well-structured, with clearly defined objectives, and the results are logically interpreted within the context of existing literature. The figures and tables are well-organized and effectively support the data; however, slight improvements in figure resolution may be necessary for publication. The discussion is comprehensive and connects the physiological and biochemical mechanisms behind the observed effects, providing a clear scientific explanation of the benefits of both treatments.
In general, the manuscript represents a valuable contribution to the field of horticulture and plant stress physiology. With only minor revisions, I believe this manuscript is suitable for publication in PeerJ.
Minor points
Line 146 the phrase "1000 cm tape" is unclear. If the authors are referring to a measuring tape or a ruler used to measure shoot length, it would be more appropriate to simply state "a measuring tape" or "a ruler."
Line 192 The phrase "10 milliliters" should be replaced with "10 mL" to conform to standard scientific conventions for units.
Line 197 The phrase "2 grams" should be replaced with "2 g" to conform to standard scientific conventions for units.

---

## Round 0.2 · accepted · Accept

· Academic Editor

Accept

Dear Dr. Abdel-Sattar

Thank you for your submission to PeerJ.

I am writing to inform you that your manuscript - 24-epibrassinosteroid and jasmine oil improve vegetative growth and productivity of Flame Seedless grapevines under abiotic stresses - has been Accepted for publication.

Congratulations!